# Aflibercept for Gastrointestinal Bleeding in Hereditary Hemorrhagic Telangiectasia: A Case Report

**DOI:** 10.3390/medicina59091533

**Published:** 2023-08-24

**Authors:** Bernat Villanueva, Adriana Iriarte, Raquel Torres-Iglesias, Miriam Muñoz Bolaño, Pau Cerdà, Antoni Riera-Mestre

**Affiliations:** 1HHT Unit, Internal Medicine Department, Hospital Universitari de Bellvitge, L’Hospitalet de Llobregat, 08907 Barcelona, Spain; bvillanueva@bellvitgehospital.cat (B.V.); adriana.iriarte@bellvitgehospital.cat (A.I.); rtorresi@bellvitgehospital.cat (R.T.-I.); pcerda@idibell.cat (P.C.); 2Bellvitge Biomedical Research Institute (IDIBELL), L’Hospitalet de Llobregat, 08908 Barcelona, Spain; 3Pharmacist Department, Hospital Universitari Bellvitge, L’Hospitalet de Llobregat, 08907 Barcelona, Spain; mmunoz@bellvitgehospital.cat; 4Faculty of Medicine and Health Sciences, Universitat de Barcelona, L’Hospitalet de Llobregat, 08907 Barcelona, Spain

**Keywords:** hereditary hemorrhagic telangiectasia, rare diseases, gastrointestinal telangiectasia, anemia, aflibercept, angiogenesis, case report

## Abstract

Herein, we present the first described hereditary hemorrhagic telangiectasia (HHT) patient treated with aflibercept for severe GI involvement after tachyphylaxis to bevacizumab, with promising results. HHT is a rare genetic disease characterized by systemic vascular malformations. Gastrointestinal telangiectasia is one of the major involvements that can produce chronic severe iron-deficiency anemia. Nowadays, support treatment with iron replacement therapy, red blood cell transfusions, and antiangiogenic drugs—mainly bevacizumab, a monoclonal antibody against vascular endothelial growth factor (VEGF)—are the main therapeutic options for this complication. The evidence of alternative drugs in patients with failure to this approach, such as tachyphylaxis to bevacizumab, is scarce. Aflibercept is a VEGF inhibitor with antiangiogenic properties approved for the treatment of different types of cancer and ocular neovascularization diseases.

## 1. Introduction

Hereditary hemorrhagic telangiectasia (HHT), also known as Rendu–Osler–Weber syndrome, is a genetic rare disease (ORPHA 774), characterized by systemic telangiectases and larger vascular malformations (VMs) [1,2]. HHT diagnosis is based on clinical criteria established by the Curaçao criteria. These criteria are (i) recurrent epistaxis, (ii) mucocutaneous telangiectasia in characteristic locations (such as lips, oral cavity, fingers, or nasal mucosa), (iii) visceral VMs, and (iv) a first-degree family member with HHT [3]. Genetic tests can also confirm the diagnosis. The most common identifiable pathogenic variants are located in endoglin (*ENG*) or activin A receptor type II-like 1 (*ACVRL1*) genes and define HHT type 1 and type 2, respectively [4,5]. *ENG* encodes endoglin, and *ACVRL1* encodes activin receptor-like kinase 1 (ALK1). Both are co-receptors at the endothelial cell surface, promoting Bone morphogenetic protein 9 (BMP9) signaling. Overall, the signaling hub BMP9/10–ENG–ALK1–Smads is essential for the angiogenic process [6]. Vascular endothelial growth factor (VEGF) is linked to BMP9/10 signaling, playing an important role in HHT pathogenesis [7]. Favorable results were found with the use of bevacizumab, an antiangiogenic monoclonal antibody against VEGF, in improving severe anemia and reducing epistaxis or gastrointestinal (GI) chronic bleeding [7,8].

GI telangiectasia is usually asymptomatic in young adults, but can provoke GI bleeding by the fifth or sixth decades [2,9]. GI bleeding is chronic and intermittent, and, because of usually accompanying epistaxis, a high clinical suspicion index is needed to diagnose this occult bleeding in HHT patients. In a recent study with HHT patients, older age, *ENG* pathogenic variants, smoking history, and hemoglobin (Hb) levels were associated with GI involvement [9]. GI bleeding can provoke severe anemia, requiring common visits to the emergency department, intravenous (IV) iron and/or red blood cell (RBC) transfusions, and reducing quality of life [9,10].

The presence of telangiectases throughout the GI tract limits the opportunity for local argon plasma coagulation (APC) therapy. This limitation usually leaves medical drugs as the only therapeutic option for HHT patients [9]. In fact, the last HHT international guidelines recommend intravenous bevacizumab or other systemic antiangiogenic drugs for those HHT patients with chronic GI bleeding who do not meet their Hb goals despite adequate iron replacement or those who require RBC transfusions [2]. However, the evidence is scarce, mainly with other therapies different from bevacizumab [11,12,13,14].

Herein, we present an HHT patient with GI involvement requiring multiple visits to the emergency department due to severe anemia despite treatment with bevacizumab. Though this situation is not very common, it is very difficult to deal with these patients with high requirements for RBC transfusions. Other options apart from bevacizumab are thalidomide, which demonstrate anti-VEGF antiangiogenic activity in HHT, but presents severe side effects, such as peripheral neuropathy; another alternative is somatostatin analogues treatment, which was described as an effective therapy for chronic GI bleeding in HHT with a good safety profile [2,14,15,16]. Our main objective is to present alternatives to bevacizumab when tachyphylaxis for treating GI bleeding occurs and to highlight this rare mechanism as a cause of loss of response to bevacizumab.

## 2. Case Report

We followed the CAse REports (CARE) guidelines to support the accuracy and usefulness of case reports [17]. Personal and clinical data collected from the patient are in line with the Spanish Data Protection Act (Ley Orgánica 3/2018 de 5 de diciembre de Protección de Datos Personales). A 63-year-old woman with HHT1 and fulfilling all Curaçao criteria was attended at our HHT Unit. This HHT Unit is the referral for adult patients from all over Catalonia (Spain), with a total population of seven million inhabitants. The patient had a history of recurrent epistaxis since childhood; multiple mucocutaneous telangiectases affecting nasal mucosa, tongue, and fingers; pulmonary arteriovenous malformations; gastrointestinal involvement; and first-degree relatives with HHT. An embolization of pulmonary arteriovenous fistulae was performed. She showed severe gastrointestinal HHT involvement, with multiple gastrointestinal telangiectases (more than 300 from the stomach to ileum, viewed through an endoscopic capsule) that caused severe anemia despite high IV iron and RBC transfusion requirements. She also received treatment with somatostatin analogues at different dosages (octreotide 50 mg twice daily, long-acting release octreotide 20 mg monthly, and long-acting release lanreotride 120 mg monthly) for a total of three years. During the last three months, treatment with lenalidomide (20 mg daily) was combined with a somatostatin analog. Moreover, endoscopic argon fulguration was performed during this period, treating thirteen 4–8 mm diameter gastroduodenal telangiectases. Despite all these therapeutic interventions, the Hb levels did not improve, and a progressive increase in RBC transfusions and IV iron requirements was observed.

She started bevacizumab at 5 mg/kg dose with an induction protocol of 4 doses every 2 weeks, 4 doses every 3 weeks, and then a once-monthly maintenance regime. She showed an improvement in anemia parameters, reducing IV iron and RBC transfusion requirements. She improved her mean Hb from 76 to 104 g/L, with a reduction in RBC unit transfusions from 27 in 1 year to 6 units in 8 months, and 12 g in 1 year to 6 g in 8 months of IV iron. After this 8-month period, the patient’s situation worsened again with a relapse in iron-deficient anemia, also adding secondary acute preserved ejection fraction heart failure symptoms, requiring intensive diuretic treatment. In this scenario, atrial fibrillation was diagnosed, and a left atrial appendage closure procedure was performed to avoid anticoagulation. There was no improvement despite increasing the bevacizumab dose to 10 mg/kg, persisting the need for visiting the emergency department with high requirements of IV iron and RBC transfusions (Figure 1).

At this point, the patient’s case was discussed in a multidisciplinary meeting and a change to an alternative antiangiogenic therapy was proposed. With the approval of the Hospital Committee of Medicines in Special Situations, aflibercept 4 mg/kg every 2 weeks was started as an off-label treatment. After 6 months, the patient presented an improvement in hemoglobin levels (mean Hb from 87 to 98 g/L) and a reduction in IV iron (from 19 g in 14 months to 4 g in 6 months) and RBC transfusion requirements (from 35 units in 14 months to 7 units in 6 months) (Figure 1). Moreover, heart failure symptoms improved, allowing a reduction in diuretic treatment and the Epistaxis Severity Score (ESS) dropped from 6.09 (moderate) to 3.26 (mild). Regarding safety, blood pressure and proteinuria were evaluated monthly and every other month, respectively, during the follow-up with no alterations and no thrombotic events reported.

## 3. Discussion

To the best of our knowledge, this is the first reported HHT patient receiving aflibercept for GI bleeding. There is a need for targeting new drugs as an alternative therapy in patients with no response to bevacizumab. In fact, loss of response or tachyphylaxis phenomena, defined as a sudden and progressive decrease in response after repetitive administration of a drug, were described in oncology and ophthalmology scenarios with antiangiogenic therapies. This situation requires switching to different anti-angiogenic drugs [18,19]. Aflibercept is a soluble VEGF decoy receptor that consists of the extracellular domains of VEGF receptors 1 and 2 and the common fraction portion of human IgG1 and it can neutralize both VEGF (A and B) and PlGF (placental growth factor). It was approved for the treatment of different types of cancer, showing higher tumor suppressive activity than bevacizumab in in vivo models (but not in patients), and for ocular neovascularization diseases (mainly age-related macular degeneration and diabetic retinopathy) [18,19]. Although the evidence in other vascular diseases is scarce, Brančíková et al. described a patient with multiple angiomatosis that presented a good response with aflibercept in a 24-month period [20]. Aflibercept has a broader spectrum and a different mechanism of action than bevacizumab, which acts as an antibody (not a VEGF trap) and its only target is VEGF-A [18]. This different mechanism could explain the improved antiangiogenic response with aflibercept observed in our patient.

Bevacizumab is recommended for patients with severe anemia, secondary to chronic GI bleeding, but there is no proposed alternative therapy when loss of response or tachyphylaxis phenomena appears. Our patient showed an initial improvement of the anemia-related parameters with bevacizumab but a successive worsening after 8 months that persisted even with a dose-increase strategy. Recently, new evidence of the use of pazopanib, a multi-target receptor tyrosine kinase inhibitor with anti-angiogenic properties, has been described in 16 HHT patients with chronic severe anemia, who previously failed to respond to other therapies, including bevacizumab therapy in half of them [21]. However, we moved to aflibercept, maintaining the same target against VEGF, because of the initial effectivity with bevacizumab. In fact, aflibercept rescued the situation in the first 6 months of treatment, achieving similar results to bevacizumab without major adverse events (Figure 1). In addition, an improvement in epistaxis and heart failure symptoms (probably related to anemia improvement) were observed. In the current scenario of personalized medicine, it is necessary to assess the potential use of biomarkers for monitoring these drugs and predicting response failures [22].

## 4. Conclusions

Aflibercept is a promising antiangiogenic therapy for HHT patients with GI bleeding and severe anemia with no response or tachyphylaxia to bevacizumab. The use of aflibercept in this scenario needs to be tested in a clinical trial.

## Figures and Tables

**Figure 1 medicina-59-01533-f001:**
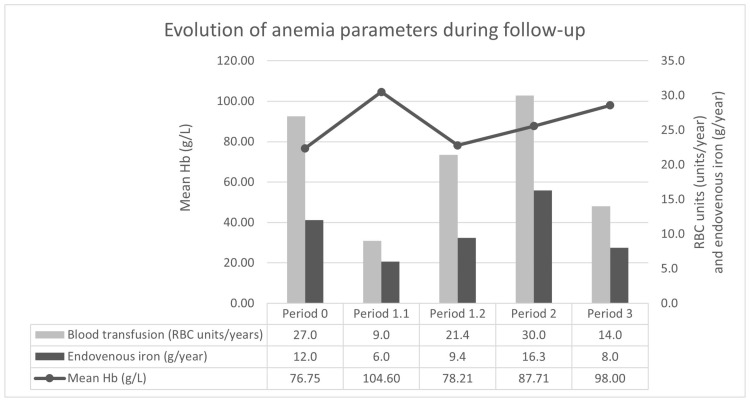
Evolution of anemia parameters during follow-up. Period 0: treatment with somatostatin analogues and lenalidomide during the year before bevacizumab started. Period 1: treatment with bevacizumab 5 mg/kg (previous induction); Period 1.1 initial period with good response (8 months); Period 1.2 last period with relapse of severe anemia (14 months). Period 2: treatment with bevacizumab 10 mg/kg monthly (14 months). Period 3: treatment with aflibercept (6 months). RBC units and IV iron were adjusted per year to allow the comparison of the results. Abbreviations: Hb, hemoglobin; RBC, red blood cells.

## Data Availability

Data is available from the corresponding authors on reasonable request.

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
