# Peer review of "Aflibercept for Gastrointestinal Bleeding in Hereditary Hemorrhagic Telangiectasia: A Case Report"

_medicina, 2023, doi:10.3390/medicina59091533_

Round 1

Reviewer 1 Report

Worldwide prevalance of Hereditary hemorrhagic telangiectasia is between 1:5000-1:10000. In guideline, bevacizumab is used as the first step. Here authors have shown that aflibercept treatment is efficient in a HHT patient who developed severe anemia despite treatment with bevacizumab. They showed that aflibercept is a promising drug but of course, one patient is not enough, more patients are required to support their results.

In my opinion the study is novel and can contribute the literature in order to reveal the effects on more patients. 

Author Response

Thank you very much for your comment and support.

Reviewer 2 Report

Dear authors the paper adresses a very interesting topic; also if presented as a rare disease HHT is a well known source of obscure gastrointestinal bleeding and treatment of sideropenic anemia in these patients is often difficult.

Among various proposed therapy in the introducion is not sufficiently detailed the role of thalidomide as antiangiogenic agent and of somatostatin analogues; these details has to be better explained in the clinical history of the patient; also in the discussion there is no indication for the use of VECF inhibitor insteadof antiangiogenic agents or somatostatin analogues; please explain more in detail; the case history should be more detailed on the endoscopic lesions an their endoscopic treatment should be discussed

Author Response

Dear authors the paper adresses a very interesting topic; also if presented as a rare disease HHT is a well known source of obscure gastrointestinal bleeding and treatment of sideropenic anemia in these patients is often difficult.

1.- Among various proposed therapy in the introducion is not sufficiently detailed the role of thalidomide as antiangiogenic agent and of somatostatin analogues; these details has to be better explained in the clinical history of the patient;

Response: Thank you very much for your comments. We added the following comment about thalidomide referring to its antiangiogenic/anti-VEGF properties in HHT:

“Other options apart from bevacizumab are thalidomide, which demonstrated anti-VEGF antiangiogenic activity in HHT, but presents severe side effects, such as peripheral neuropathy; or somatostatin analogues treatment, which has been described an effective therapy for chronic GI bleeding in HHT, with a good safety profile [2,14-16].”

We also have added more detailed information about the pre-bevacizumab therapies of the patient in her clinical history report with specific drugs, posology and response:

“She also received treatment with somatostatin analogues at different dosages (octreotide 50 mg twice daily, long acting release octreotide 20 mg monthly and long acting release lanreotride 120 mg monthly) for a total of three years. During the last three months, treatment with lenalidomide (20mg daily) was combined with a somatostatin analog. Moreover, endoscopic argon fulguration was performed during this period, treating thirteen 4-8 mm diameter gastroduodenal telangiectases. Despite all these therapeutic interventions, Hb levels did not improve and a progressive increase of RBC transfusions and IV iron requirements were observed.”

2.- also in the discussion there is no indication for the use of VECF inhibitor instead of antiangiogenic agents or somatostatin analogues; please explain more in detail;

Response: We probably are not fully understanding which explanation is asked in this point (“VECF” inhibitor?). In the Discussion section we already compare the antiangiogenic situation in patients with cancer and those in ophthalmology scenario, where sequencing antiangiogenic drugs is needed. Though there are some reports that suggest the anti-VEGF effect with somatostatin analogues, there isn’t enough evidence to compare this treatment with bevacizumab, aflibercept or pazopanib, regarding the antiangiogenic effect.

3.- the case history should be more detailed on the endoscopic lesions and their endoscopic treatment should be discussed

Response: We have detailed the pre-bevacizumab endoscopic treatment as you recommended and previously explained (please, see the first response).

Reviewer 3 Report

1. I would suggest to change the title from "Aflibercept for gastrointestinal bleeding in hereditary hemorrhagic telangiectasia: modifying the drug but not the target" to "Aflibercept for gastrointestinal bleeding in hereditary hemorrhagic telangiectasia: case report"

2. Abstract. Please move the last sentence in the abstract (Herein, we present...) to the beginning

Author Response

Thank you very much for your comments. We have modified the Title and the Abstract according to both reviewer's comments.

Reviewer 4 Report

Comments and revisions:

Guideline:
Present your study per The CARE guidelines (for CAse REports).

Introduction and Background:
The introduction provides a concise overview of hereditary hemorrhagic telangiectasia (HHT), its diagnosis, and the role of VEGF in HHT pathogenesis. However, it lacks a clear statement of the research question or the specific objective of the case report. It would be beneficial to clearly state the purpose of the case report at the beginning to provide readers with a better understanding of the study's scope.

Revision: Include a clear statement of the research question or the specific objective of the case report in the introduction.

Case Presentation:
The case presentation provides relevant clinical information about the patient's HHT diagnosis and the treatments she received. However, the description of the treatments could be more cohesive and could benefit from a more organized and structured presentation of the patient's treatment history. Additionally, it would be helpful to include a timeline of the treatments and their outcomes to provide a better understanding of the patient's clinical course.

Revision: Organize the case presentation with a clear timeline of the treatments and their outcomes to enhance clarity and coherence.

Discussion:
The discussion provides valuable insights into using bevacizumab and the subsequent switch to aflibercept in managing GI bleeding in HHT. However, including a more comprehensive discussion of the reasons for the patient's partial response to bevacizumab and the potential mechanisms underlying the improved response to aflibercept would be helpful. The authors could explore the differences between the two drugs and their modes of action to shed light on why one might be more effective than the other in this specific case.

Revision: Expand the discussion to include a comparative analysis of bevacizumab and aflibercept, discussing their mechanisms of action and potential reasons for the differential response observed in the patient.

Limitations and Future Directions:
The case report would benefit from a section discussing the limitations of the study. Since this is a single-patient case report, the findings may not be generalizable to the broader HHT population. Additionally, a discussion of potential future research directions, such as exploring other antiangiogenic therapies or combination treatments, could be included to guide further investigations in this area.

Revision: Add a section on limitations and future directions to acknowledge the constraints of a single-patient case report and suggest potential avenues for future research.

Minor editing of English language is required.

Author Response

Guideline:
Present your study per The CARE guidelines (for CAse REports).

Response: Thank you very much for your comment. We have followed the CAse REports (CARE) guidelines to support the accuracy and usefulness of our case report.

Introduction and Background:
The introduction provides a concise overview of hereditary hemorrhagic telangiectasia (HHT), its diagnosis, and the role of VEGF in HHT pathogenesis. However, it lacks a clear statement of the research question or the specific objective of the case report. It would be beneficial to clearly state the purpose of the case report at the beginning to provide readers with a better understanding of the study's scope.

Revision: Include a clear statement of the research question or the specific objective of the case report in the introduction.

Response: Many thanks. A main objective for the case report has been added as follows:

“Our main objective is to present alternatives to bevacizumab when tachyphylaxis for treating GI bleeding occurs and to highlight this rare mechanism as a cause of loss of response to bevacizumab”.

Case Presentation:
The case presentation provides relevant clinical information about the patient's HHT diagnosis and the treatments she received. However, the description of the treatments could be more cohesive and could benefit from a more organized and structured presentation of the patient's treatment history. Additionally, it would be helpful to include a timeline of the treatments and their outcomes to provide a better understanding of the patient's clinical course.

Revision: Organize the case presentation with a clear timeline of the treatments and their outcomes to enhance clarity and coherence.

Response: Thank you very much for your comment. We have rewriten the case report adding a timeline of the treatments administered:

“She also received treatment with somatostatin analogues at different dosages (octreotide 50 mg twice daily, long acting release octreotide 20 mg monthly and long acting release lanreotride 120 mg monthly) for a total of three years. During the last three months, treatment with lenalidomide (20mg daily) was combined with a somatostatin analog. Moreover, endoscopic argon fulguration was performed during this period, treating thirteen 4-8 mm diameter gastroduodenal telangiectases. Despite all these therapeutic interventions, Hb levels did not improve and a progressive increase of RBC transfusions and IV iron requirements were observed.”

Discussion:
The discussion provides valuable insights into using bevacizumab and the subsequent switch to aflibercept in managing GI bleeding in HHT. However, including a more comprehensive discussion of the reasons for the patient's partial response to bevacizumab and the potential mechanisms underlying the improved response to aflibercept would be helpful. The authors could explore the differences between the two drugs and their modes of action to shed light on why one might be more effective than the other in this specific case.

Revision: Expand the discussion to include a comparative analysis of bevacizumab and aflibercept, discussing their mechanisms of action and potential reasons for the differential response observed in the patient.

Response: Thanks a lot. A little explanation about potential increased action with aflibercept was added:

“Aflibercept has a broader spectrum and a different mechanism of action than bevacizumab, which acts as an antibody (not a VEGF trap) and its only target is VEGF-A [18]. This different mechanism could explain the improved antiangiogenic response with aflibercept observed in our patient”.

Limitations and Future Directions:
The case report would benefit from a section discussing the limitations of the study. Since this is a single-patient case report, the findings may not be generalizable to the broader HHT population. Additionally, a discussion of potential future research directions, such as exploring other antiangiogenic therapies or combination treatments, could be included to guide further investigations in this area.

Revision: Add a section on limitations and future directions to acknowledge the constraints of a single-patient case report and suggest potential avenues for future research.

Response: The authors are aware of the limitations of a case report. For this reason, one of our conclusions is that a clinical trial is needed to generalize the use of aflibercept in HHT patients with GI involvement. Many thanks for the comment. However, we think that this limitationis already established and we had already highlighted the next steps to perfom.

Round 2

Reviewer 4 Report

Thank you for your revisions.

Minor editing of English language required.